# A Long-Term Incidence of Heart Failure and Predictors Following Newly Developed Acute Myocardial Infarction: A 10 Years Retrospective Cohort Study with Korean National Health Insurance Data

**DOI:** 10.3390/ijerph18126207

**Published:** 2021-06-08

**Authors:** Hyojung Choi, Joo Yeon Seo, Jinho Shin, Bo Youl Choi, Yu-Mi Kim

**Affiliations:** 1Health Insurance Review and Assessment Service, Wonju 26465, Korea; choihyojung@hira.or.kr; 2Department of Preventive Medicine, College of Medicine, Hanyang University, Seoul 04763, Korea; rirooroo@gmail.com (J.Y.S.); bychoi@hanyang.ac.kr (B.Y.C.); 3Division of Cardiology, Department of Internal Medicine, College of Medicine, Hanyang University, Seoul 04763, Korea; jhs2003@hanyang.ac.kr; 4School of Public Health, Hanyang University, Seoul 04763, Korea

**Keywords:** heart failure, myocardial infarction, incidence, prognosis, risk factors

## Abstract

Heart failure (HF) is the major mechanism of mortality in acute myocardial infarction (AMI) during early or intermediate post-AMI period. But heart failure is one of the most common long-term complications of AMI. Applied the retrospective cohort study design with nation representative population data, this study traced the incidence of late-onset heart failure since 1 year after newly developed acute myocardial infarction and assessed its risk factors. **Methods and Results:** Using the Korea National Health Insurance database, 18,328 newly developed AMI patients aged 40 years or older and first hospitalized in 2010 for 3 days or more, were set up as baseline cohort (12,403). The incidence rate of AMI per 100,000 persons was 79.8 overall, and 49.6 for women and 112.3 for men. A total of 2010 (1073 men, 937 women) were newly developed with HF during 6 years following post AMI. Cumulative incidences of HF per 1000 AMI patients for a year at each time period were 37.4 in initial hospitalization, 32.3 in 1 year after discharge, and 8.9 in 1–6 years. The overall and age-specific incidence rates of HF were higher in women than men. For late-onset HF, female, medical aid, pre-existing hypertension, severity of AMI, duration of hospital stay during index admission, reperfusion treatment, and drug prescription pattern including diuretics, affected the occurrence of late-onset HF. **Conclusion:** With respect to late-onset HF following AMI, appropriate management including hypertension and medical aid program in addition to quality improvement of AMI treatment are required to reduce the risk of late-onset heart failure.

## 1. Introduction

Heart failure (HF) is clinically defined as a syndrome by a series of symptoms and signs produced by complex circulatory and neurohormonal responses [1]. HF usually causes the burden of hospitalization, and its prevalence continues to increase due to population aging and an increase in comorbidities, such as hypertension and diabetes [2]. In the Republic of Korea, prevalence of HF is expected to double from 1.6% in 2015 to 3.4% in 2040 [3].

Acute myocardial infarction (AMI) is known as the main antecedent cause of HF, and there are plenty of studies established the short-term HF occurrence complicating AMI [4,5,6]. Previous studies have shown that age, being male, smoking, diabetes, high blood pressure, a history of stroke, atrial fibrillation or cardiogenic shock, not reperfusion within 24 h, and long length of hospital stays at the time of index AMI were the major risk factors for heart failure after myocardial infarction [4,6]. In addition, considering that HF complicating post AMI is predictor of mortality [7] and the AMI case fatality rates decrease due to improved reperfusion therapy, the incidence and risk factors for prognosis of HF post MI would be imperative for the appraisal of HF epidemics [6].

HF post AMI could be considered an indicator of the quality of acute phase care following AMI, and the management of long-term sequelae is becoming increasingly important. There is a paucity of epidemiologic evidence on long-term prognosis and risk factors for HF post AMI, especially in the East-Asian population.

Applying the retrospective cohort study design with nation representative population data, this study traced the prognosis of HF complicating newly developed AMI, and assessed the impact of patient characteristics and healthcare use on the occurrence of HF in patients who survived AMI during the acute or subacute phase.

## 2. Methods

### 2.1. Data Sources and Study Subjects

This study used customized health information data from the Korean NHI database. The NHI Corporation provides health insurance claims data, including information related to healthcare use, treatment, and all paid services provided to patients (such as diagnosis, examination, procedures, and drugs). As NHI covers about 99% of all Korean citizens, it can be viewed as a representative source of data. Using this population based national representative large database, a retrospective cohort study was designed.

This study conducted in accordance with World Medical Association Declaration of Helsinki. The study protocol was approved by the institutional review board (IRB) of Hanyang University (IRB No. HYI-17-115-1).

### 2.2. Definition of First Episode of AMI and Baseline Cohort

For the diagnosis of first episode of AMI (ICD-10 code: I21.x), an operational definition with no past history of AMI for 3 years from 2007 to 2009 was applied. Therefore, the patients who had been hospitalized for AMI between 1 January 2007 and 31 December 2009, and who had been detected in primary clinics, were excluded.

Finally, the baseline cohort was constructed to include the patients aged 40 years or older and first hospitalized for AMI in 2010 for 3 days or more (index hospitalization). Among the 18,328 baseline cohort of newly developed AMI, 12,403 (67.7%) were men and 5925 (32.3%) were women.

### 2.3. Follow-Up and HF Occurrence

Among the selected patients, those who had not been diagnosed with or treated for HF (ICD-10 code: I50.x) up to the onset of AMI were monitored for 6 years after index hospitalization. A late-onset HF was defined among patients who survived over 1 year after initial hospitalization.

### 2.4. The Risk Factors on Late-Onset HF Post AMI

This study assessed the characteristics of patients at the time of diagnosis (baseline cohort) and the effect of medical interventions during the 6 months after the diagnosis of AMI.

AMIs were categorized as ST-elevation myocardial infarction (STEMI), non-ST elevation myocardial infarction (NSTEMI), and unspecified AMI (UAMI). Patients who underwent coronary artery bypass graft (CABG) surgery, or were administered intravenous diuretics during index admission period, were classified severe patients. The remaining patients were classified as moderate patients.

Healthcare use was characterized by the type of medical institution in which index hospitalization occurred (hospital, general hospital, or tertiary hospital), duration of hospital stay (days), type of revascularization therapy (thrombolytics, Percutaneous Coronary Intervention (PCI), and/or CABG), and medication. Revascularization therapy including thrombolytics, PCI, and CABG were categorized in order according to the level of treatment, and when two or more treatments were performed, a higher level of treatment was selected. Drug treatment was characterized by the number of drugs used, the use of diuretics, and the use of 4 types of drugs (antiplatelet agent, cholesterol-lowering drug, RAS inhibitor, beta blocker) recommended in the guidelines for the treatment of AMI.

### 2.5. Statistical Analysis

The age specific incidence rates for newly developed AMI in 2010 were calculated as the cases defined as the first episode of AMI from NHI database, divided by the number of population data from Statistics Korea per 100,000 population.

The cumulative incidence and case-fatality rate of HF per 1000 AMI patients were calculated at initial hospitalization, 1 year after discharge, and 2–6 years after discharge.

The variables of age and duration of hospital stay in the index hospitalization period were considered as continuous variables, and other risk factors as categorical variables. Categorical variables are presented as the number of patients and proportion (%), and continuous variables are presented as mean and standard deviation. Hazard ratios (HRs) were calculated using a Cox’s proportional hazards model to identify factors affecting the development of HF after the onset of AMI. The first hospitalization for HF during the follow-up period was regarded as an event. For patients who did not have HF, the time of death was regarded as censored if they died, and in the case of no death, the time of 6 years from the index admission discharge date was considered to be censored. SAS 9.4 (SAS Institute, Cary, NC, USA) was used for all statistical analyses, and the significance level was set at 0.05.

## 3. Results

### 3.1. Incidence Rate of Newly Developed AMI

From 1 January 2010 to 31 December 2010, 18,328 patients over 40 years of age in Korea newly developed AMI. Among them, 12,403 (67.7%) were men and 5925 (32.3%) were women. The incidence rate of AMI per 100,000 persons was 79.8 overall, and 49.6 for women and 112.3 for men (Table 1). In all age groups, the incidence in men was higher than that in women; the incidence in men increased sharply in those aged 50 years and older, while the incidence in women increased significantly from the age of 60 years onwards. Both men and women showed the highest incidence of AMI in the age group over 80 years of age.

Demographic characteristics and pre-existing conditions of newly developed AMI patients in 2010 were summarized in Appendix A.

### 3.2. Cumulative Incidence Rates of HF and Case-Fatality Rate Among Newly Developed AMI during 6 Years Follow Up

In 2010, 18,328 patients were initially diagnosed for AMI, of whom 2010 (1073 men, 937 women) were diagnosed with newly developed HF during the 6 years following their first episode of AMI (Table 2). A total of 685 (353 men, 332 women) was diagnosed with HF at initial admission (37.4 per 1000 AMI patients for a year). Cumulative incidences of HF per 1000 AMI patients for a year at each time periods were 32.3 in 1 year after discharge, and 8.9 in 1–6 years. The overall incidence rate of HF during observational period was higher in women (31.3 per 1000 AMI patients for a year) than men (15.8 per 1000 AMI patients for a year). All age-specific incidence rates were higher in women than men, and it was more clearly observed after the 50s. Overall, the incidence of HF during index hospitalization was highest, the incidence is high until the next year, and the incidence rate is lowest during the following 5 years.

The number of deaths which could be detected only by medical care use information was 5217 (Table 3). Case fatality rate for newly developed AMI at each time periods were 85.0 per 1000 AMI patients in initial hospitalization, 52.2 in 6 months after discharge, 22.8 in 7–12 months after discharge, and 33.9 in 1–6 years after discharge. Among the death cases in the index hospitalization, the HF was diagnosed as 8.1%, 9.6% of the deaths during 6 months after initial discharge were HF, and 4.4% of those during 7–12 months after initial discharge. during 1–6 years after initial discharge, 11.9% of deaths were diagnosed with HF.

### 3.3. The Impacts of Characteristics, Health Care Utilization and Medication in Index Hospitalization on Late Onset HF Incidence

Patients who survived over 1 year after initial hospitalization were defined as a late-onset HF. Table 4 shows the effects of the characteristics including healthcare use and medication on the development of HF after 1 year following newly developed AMI, examined using Cox’s proportional hazards models. The risk of late-onset HF increased by 1.046 times (95% CI: 1.039–1.054) as age increased. The risk in patients with hypertension was 1.242 times higher (95% CI: 1.041–0.482) than that in patients without hypertension. Higher severity AMI showed 2.487 times greater HR (95% CI: 2.105–2.938). The risk of HF was 1.001 times higher (95% CI: 1.000–1.002) in individuals with longer durations of hospital stay. The patients who were treated by PCI and/or CABG resulted in lower occurrence of late-onset HF (HR: 0.627, 95% CI: 0.515–0.764 for PCI, and HR: 0.435, 95% CI: 0.271–0.697 for CABG). The risk of HF in those who used diuretics at index admission was 1.809 times higher than that among those who did not use these drugs (95% CI: 1.499–2.185).

Reperfusion therapies, including CABG, PCI, and Thrombolytics, were categorized in order according to the level of treatment; when two or more treatments were performed, the higher level of treatment was selected. Therefore, each patient was treated independently.

## 4. Discussion

This study traced the prognosis of newly developed AMI as HF occurrence and case-fatality rate for 6 years, and assessed the risk factors on late-onset HF occurrence following AMI using the retrospective cohort study design with nation representative population based on NHI data.

The number of patients newly diagnosed with AMI in 2010 was 18,328 in this study, representing about 0.04% of the population. As of 2010, the incidence rate (IR) of AMI in those over the age of 40 years in Korea was 79.8 per 100,000 persons. The IR was higher in men (112.3 per 100,000) than in women (49.6 per 100,000). Both men and women showed the highest IR of AMI in the age group over 80 years of age. The Korea Disease Control and Prevention Agency reported that the incidence of angina pectoris and myocardial infarction among those aged >30 years had increased sharply from 0.8% in 1998 to 2.4% in 2013 [8].

In Hong and Kang’s study, the incidence of AMI in 2007 was 91.8 per 100,000 population for all age groups [9], whereas in another Korean study, the IR of AMI in 2010 was 29.4 per 100,000 population including all age groups [10]. An interpretation might be that the IR of AMI decreased gradually over time; however, these findings could also be a result of the difference across the studies in terms of study population, operational disease definition, and study conduction and analysis. In most studies in countries outside Korea, the IR tended to decrease gradually over time [11]. Nevertheless, since acute diseases such as AMI place a large burden on the country and individual patients, it is necessary to establish a system to monitor diseases in the long term through sufficient research. Therefore, it becomes necessary to examine in-depth changes in the IR of AMI in the long term, based on representative data such as health insurance claim data, in the future.

A total of 2010 (1073 men, 937 women) were newly developed with HF during the 6 years following AMI. The cumulative incidence rate (CIR) of HF after the first onset of AMI, was 123.2 per 1000 AMI patients for 6 years, and 20.5 per 1 year. CIR of HF per 1000 AMI patients for a year at each time periods were 37.4 in initial hospitalization, 32.3 in 1 year after discharge, and 8.9 in 1–6 years. The longer the follow-up period, the lower the HF incidence; the incidence decreased between 2–5 years. The reported incidence of HF after the onset of AMI in previous studies differed (9.9% [12], 11.2% [13], 11.3% [14], and 13.2% [15]) depending on subjects and follow-up period, but the trend was similar to that noted in the present study, with a gradual decrease over time [16]. As patients who had suffered from HF before AMI, but did not receive medical services for the condition might be included in this study, the calculated CIR could be overestimated.

The average age of patients who newly developed HF post AMI was 67.1 years in men and 75.4 years in women. In several non-Korean studies, the average age of such patients varied from 62 to 75 years [4,17,18]. The overall incidence rate of HF during observational period was higher in women (31.3 per 1000 AMI patients for a year) than men (45.8 per 1000 AMI patients for a year) and all age-specific incidence rates were higher in women than men. The study reported that in Australia, the incidence of HF among those with a previous AMI was 16.5% [19]. In a study based on administrative data from the United States, the incidence of HF within 1 year following the onset of myocardial infarction was reported to be 14.6% [20]. A study based on national myocardial infarction registration data from the United States showed that the incidence of HF during hospitalization was approximately 19.1%, and a 5-year follow-up study showed that HF occurred in about 6.3% of patients with reported myocardial infarction [16]. A Korean study based on health insurance claims data examined the complications that occur after AMI; it showed that 35.2% of congestive HF and left ventricular failure cases occurred in patients with AMI who had coronary artery disease and were thus in a high-risk group [21]. In a study conducted on patients included in the Korea AMI Registry, approximately 18.1% were found to develop total HF [22]. Nevertheless, it is difficult to make clear and direct comparisons, given the differences in patient characteristics and observation period across studies. The following were known as factors affecting the occurrence of HF following AMI: presence of diabetes or hypertension, and previous MI [19]. The proportion of HF cases was higher among patients with underlying diseases, except among those with dyslipidemia. These results are consistent with previous studies that reported a higher incidence of HF among patients with hypertension and diabetes [18,23].

Considering sex difference, the incidence of HF following AMI was higher in men and older individuals [20], and women died during hospitalization more often than men in another study [16]. In this study, the incidence density of patients who developed HF was highest among patients who underwent CABG surgery in women, and higher among patients who were prescribed diuretics in women.

In addition, the incidence density of HF was highest among those admitted to a tertiary hospital, those with a long hospital stay duration, those who were prescribed thrombolytics in men, those who underwent CABG in women during the index admission, and those who were prescribed no recommended drugs. This may indicate whether proper prescription according to the treatment guidelines is important for the prevention of HF. In previous studies, the average length of hospital stay in patients who developed HF after AMI was 8.1 days, compared to 6.8 days in patients who did not develop HF [16]. In a previous study, more patients who developed HF were treated with reperfusion therapy for AMI [4,17]. However, another study showed that the use of PCI for AMI in patients who subsequently developed HF was less common than that in patients who did not develop HF, but the rate of CABG was higher in the former group of patients [24]. Another study showed that the rate of CABG use was higher in patients who did not develop HF [16]. According to the results of the present study, the incidence density of HF in the group using none of the recommended drugs was highest, and the incidence densities in the group using only one drug in men and three drugs in women were the second highest at index admission. Regarding diuretics, the incidence densities of HF in the group using diuretics were markedly higher than that in the group of patients who did not use this medication in both men and women, not only at index admission, but at the point 6 month after index admission.

We identified the risk of sociodemographic characteristics, underlying diseases at the time of AMI diagnosis, types of AMI (STEMI, NSTEMI, and UAMI), severity, and the characteristics of healthcare use (type of hospital, duration of hospital stay, type of revascularization therapy, and medication) of late-onset HF post AMI with Cox’s proportional hazards model. As a result, women, higher age, hypertension patients, higher severity, long hospital stay, use of diuretics at the time of diagnosis, and prescription of three or more recommended drugs for 6 months after index admission were found to increase the risk of developing HF. In contrast, the risk of HF was lower among the patients with health insurance, those who underwent PCI or CABG at the index admission, and those who used two of the recommended drugs at index admission, as compared to those receiving medical treatment. Other studies also showed that older patients, women, those with diabetes, and those who underwent PCI or CABG procedures had a higher incidence of HF [17]. In Korea, the relationship between the occurrence of HF and the patient’s healthcare use characteristics had not been elucidated to date, and thus this study provides an important epidemiologic evidence.

This study had several limitations. It is possible that some patient’s healthcare usage information may have been omitted as non-insurance details were not included. In addition, as the health insurance claim data include only patients who were treated at a medical institution, the results may have been underestimated due to omission of untreated patients. Furthermore, patients with underlying diseases, such as diabetes, hypertension, and dyslipidemia, but without medical use, could have been excluded from calculations, leading to underestimation. In addition, education level, and health behaviors such as smoking and drinking, were not included in the analysis. Moreover, the indications for treatment and prescription were not considered as clinical information on the results of echocardiography or blood tests was not included. In particular, analyses including clinical factors such as the incidence rate of heart failure according to the location and size of the infarct site, cardiac enzyme level, whether the case was a left or right heart failure, and echocardiography patterns are needed. Therefore, in the future, we intend to conduct research by constructing data including such clinical information. This population-based retrospective cohort study, performed using 10 years of Korean Health Insurance (NHI) data (2007–2016), provides representative data and a basis for establishing appropriate preventive plans depending on the characteristics of patients with HF occurrence in chronic phase following first episode of AMI.

In conclusion, the incidence of HF post AMI was most common at the index hospitalization and decreased as time passed, but it steadily occurred between 1–6 years, and there were also deaths. Therefore, investigating the characteristics of the late-onset HF post AMI has an important health significance in that it can reduce the incidence through prevention by identifying high-risk groups. We found that sex, type of medical insurance, severity, duration of hospital stay (days) at index admission, and drug prescription and treatment for AMI had an effect on late-onset HF among 1 year survivors after first episode of AMI. Therefore, in order to reduce the risk of HF occurrence in chronic phase, appropriate prescriptions and treatment are required. In the case of older patients with high blood pressure and women patients who showed higher CIR of HF post AMI, more advanced management is necessary. Furthermore, second and third prevention efforts for patients with a high risk of HF are needed to improve long-term prognosis.

## Figures and Tables

**Table 1 ijerph-18-06207-t001:** Incidence rate of newly diagnosed AMI according to age and sex in 2010.

Age	Total			Men			Women		
	N	AMI	IR	N	AMI	IR	N	AMI	IR
40–49	8,715,639	2059	(23.6)	4,434,091	1921	(43.3)	4,281,548	138	(3.2)
50–59	6,726,716	3918	(58.2)	3,369,420	3432	(101.9)	3,357,296	486	(14.5)
60–69	4,092,725	4547	(111.1)	1,946,828	3299	(169.5)	2,145,897	1248	(58.2)
70–79	2,536,642	5066	(199.7)	1,027,907	2765	(269.0)	1,508,735	2301	(152.5)
80+	907,282	2738	(301.8)	261,387	986	(377.2)	645,895	1752	(271.3)
Total	22,979,004	18,328	(79.8)	11,039,633	12,403	(112.3)	11,939,371	5925	(49.6)

N: number, IR: incidence rate per 100,000 persons.

**Table 2 ijerph-18-06207-t002:** Cumulative incidence of HF among newly developed AMI during 6 years follow-up according to sex and age.

Sex	Age	AMI	Heart Failure							
			Index Hospitalization		Until 1 Year after Discharge		During 1–6 Years		Overall	
		N	N	CIR	N	CIR	N	CIR	N	CIR
Men	40–49	1921	26	13.5	32	16.9	38	4.1	96	8.8
	50–59	3432	62	18.1	55	16.3	76	4.6	193	9.9
	60–69	3299	82	24.9	61	19.0	114	7.2	257	14.1
	70–79	2765	112	40.5	92	34.7	149	11.6	353	24.4
	80+	986	71	72.0	46	50.3	57	13.1	174	35.7
	Total	12,403	353	28.5	286	23.7	434	7.4	1073	15.8
Women	40–49	138	4	29.0	3	22.4	3	4.6	10	13.0
	50–59	486	14	28.8	15	31.8	18	7.9	47	17.8
	60–69	1248	59	47.3	39	32.8	48	8.3	146	22.1
	70–79	2301	125	54.3	107	49.2	145	14.0	377	32.7
	80+	1752	130	74.2	119	73.4	108	14.4	357	42.7
	Total	5925	332	56.0	283	50.6	322	12.1	937	31.3
Total	40–49	2059	30	14.6	35	17.2	41	4.1	106	9.0
	50–59	3918	76	19.4	70	18.2	94	5.0	240	10.9
	60–69	4547	141	31.0	100	22.7	162	7.5	403	16.2
	70–79	5066	237	46.8	199	41.2	294	12.7	730	28.1
	80+	2738	201	73.4	165	65.0	165	13.9	531	40.1
	Total	18,328	685	37.4	569	32.3	756	8.9	2010	20.5

N: number, CIR: cumulative incidence rate per 1000 AMI patients for a year.

**Table 3 ijerph-18-06207-t003:** Case-fatality rates following AMI according to observation periods.

Period	Death	Heart Failure			Total	
		No	Yes			
		N	N	CIR	N	CFR
Index hospitalization	No	16,211	559	33.3	16,770	
	Yes	1432	126	80.9	1558	85.0
	Total	17,643	685	37.4	18,328	
6 months after initial discharge	No	15,067	298	19.4	15,365	
	Yes	765	81	95.7	846	52.2
	Total	15,832	379	23.4	16,211	
7–12 months after initial discharge	No	14,558	175	11.9	14,733	
	Yes	328	15	43.7	343	22.8
	Total	14,886	190	12.6	15,076	
1–6 years after initial discharge	No	11,625	463	38.3	12,088	
	Yes	2177	293	118.6	2470	33.9
	Total	13,802	756	51.6	14,558	

N: number, CIR: cumulative incidence per 1000 AMI patients for a year, CFR: case fatality rate per 1000 AMI patients for a year.

**Table 4 ijerph-18-06207-t004:** The impacts of covariates on late onset HF with Cox-proportional hazard models.

	HR	95% CI	*p* Value
**Demographic and clinical characteristics**			
**Sex**			
Men (ref.)			
Women	0.954	(0.813–1.119)	0.5603
**Age (years)**	1.046	(1.039–1.054)	<0.0001
**Type of medical insurance**			
Medical aid (ref.)			
NHI	0.554	(0.453–0.679)	<0.0001
**Pre-existing conditions**			
No diabetes (ref.)			
Diabetes	1.126	(0.968–1.310)	0.1252
No hypertension (ref.)			
Hypertension	1.242	(1.041–1.482)	0.0159
No dyslipidemia (ref.)			
Dyslipidemia	0.873	(0.744–1.026)	0.0988
**Type of AMI**			
STEMI (ref.)			
NSTEMI	1.074	(0.846–1.363)	0.5586
Unspecified	1.109	(0.932–1.318)	0.2434
Severity			
Moderate (ref.)			
Severe	2.487	(2.105–2.938)	<0.0001
***Health care utilization***			
**Hospital size**			
Tertiary general hospital (ref.)			
General hospital	0.998	(0.860–1.157)	0.9779
Hospital	1.321	(0.880–1.982)	0.1789
**Length of stay (days)**	1.001	(1.000–1.002)	0.0126
**Reperfusion therapy**			
No (ref.)			
Thrombolytics	1.401	(0.619–3.175)	0.4187
PCI	0.627	(0.515–0.764)	<0.0001
CABG	0.435	(0.271–0.697)	0.0005
***Medication***			
**Number of drugs**			
0 (reference)			
1–2	0.597	(0.411–0.866)	0.0065
3+	0.769	(0.553–1.069)	0.1181
**Diuretic**			
No (reference)			
Yes	1.809	(1.499–2.185)	<0.0001

HR: Hazard Ratio, CI: Confidence Interval, Ref: reference, NHI: National Health Insurance, STEMI: ST segment Elevation Myocardial Infarction, NSTEMI: non-ST segment Elevation Myocardial Infarction, and PCI: Percutaneous Coronary Intervention, CABG: Coronary Artery Bypass Gift.

## Data Availability

The data that support the findings of this study are available from Korea National Health Insurance Service but restrictions apply to the availability of these data, which were used under license for the current study, and so are not publicly available.

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
