# Peer review of "A Long-Term Incidence of Heart Failure and Predictors Following Newly Developed Acute Myocardial Infarction: A 10 Years Retrospective Cohort Study with Korean National Health Insurance Data"

_ijerph, 2021, doi:10.3390/ijerph18126207_

Round 1

Reviewer 1 Report

This is a very interesting study from South Korea, looking at the incidence of heart failure following myocardial infarction. If I understand correctly, then NSTEMI and STEMI patients are mixed. Overall there is certainly merit in the data, but some aspects are missing in order to make it more meaningful for the readership. The readers would like to know more about the real-life risk of certain patient populations. As an example, it seems obvious how gender and age affect the diagnosis (AMI, HF), but not the extent of infarction itself:

  1. We need to completely separate NSTEMI / STEMI
  2. It is important to know whether STEMI patients were reperfused 
  3. It is important to know sth about infarct size in STEMI. A small troponin T rise is meaningless, where as hs cTnT of over 5000 is typically associated with larger infarcts. Alternatively echocardiogram before discharge would be helpful as well
  4. Introduction is not good. We would like to know which factors actually affect heart failure post MI, e.g. how about inflammation ?
  5. How can we compare the STEMI/NSTEMI patients; is there a "healthy age-matched" population, maybe all its with hypertension alone but no heart attack. Or at least use NSTEMI patients as comparison head to head.

Reviewer 2 Report

  1. In page3 Tab1,The word that expresses the number of people cannot use “NO”.
  2. What are the selection criteria for patients with heart failure? Left heart failure or right heart failure? There is clear criteria for left heart failure. But there is no clinically quantifiable standard for right heart failure
  3. The probability of complications of heart failure caused by acute myocardial infarction in different parts is different. It will be more accurate to separate the myocardial infarction from different parts in the statistics.
  4. The topic is too complicated and the topic does not correspond to the conclusion.

Round 2

Reviewer 2 Report

The authors addressed sufficiently my comments and the manuscript has been improved.